# Quality of Life in Presbyopes with Low and High Myopia Using Single-Vision and Progressive-Lens Correction

**DOI:** 10.3390/jcm10081589

**Published:** 2021-04-09

**Authors:** Adeline Yang, Si Ying Lim, Yee Ling Wong, Anna Yeo, Narayanan Rajeev, Björn Drobe

**Affiliations:** 1Essilor R&D AMERA, Essilor International, Singapore 339346, Singapore; yeeling.wong@essilor.com.sg (Y.L.W.); drobeb@essilor.com (B.D.); 2School of Chemical & Life Sciences, Singapore Polytechnic, Singapore 139651, Singapore; siying100@hotmail.com (S.Y.L.); Narayanan_RAJEEV@sp.edu.sg (N.R.); 3Education & Professional Services, Essilor AMERA, Singapore 339338, Singapore; anna.yeo@essilor.com.sg

**Keywords:** myopia, quality of life, RSVP questionnaire, presbyopia

## Abstract

This study evaluates the impact of the severity of myopia and the type of visual correction in presbyopia on vision-related quality of life (QOL), using the refractive status and vision profile (RSVP) questionnaire. A total of 149 subjects aged 41–75 years with myopic presbyopia were recruited: 108 had low myopia and 41 had high myopia. The RSVP questionnaire was administered. Rasch analysis was performed on five subscales: perception, expectation, functionality, symptoms, and problems with glasses. Highly myopic subjects had a significantly lower mean QOL score (51.65), compared to low myopes (65.24) (*p* < 0.001). They also had a significantly lower functionality score with glasses (49.38), compared to low myopes (57.00) (*p* = 0.018), and they had a worse functionality score without glasses (29.12), compared to low myopes (36.24) (*p* = 0.045). Those who wore progressive addition lenses (PAL) in the high-myope group (*n* = 25) scored significantly better, compared to those who wore single-vision distance (SVD) lenses (*n* = 14), with perception scores of 61.19 and 46.94, respectively (*p* = 0.029). Highly myopic presbyopes had worse overall QOL and functionality, both with and without glasses, compared to presbyopes with low myopia. High-myopic PAL users had a better perception outcome than SVD lens wearers. Low-myopic PAL wearers had a better QOL than SVD wearers.

## 1. Introduction

Presbyopia is a global problem affecting 1.8 billion people worldwide [1], of which at least 826 million were not adequately corrected as of 2015 [2]. The number of presbyopes is set to increase to 2.1 billion by 2030, against a backdrop of an ageing global population where the median age could reach 40 years by 2050 [3]. While the impact of presbyopia can be minimised easily by using visual correction such as spectacles, contact lenses, or refractive surgery, up to 34% of presbyopes in developed countries do not have adequate correction [4]. This is compounded by the projected rise in myopia’s prevalence and severity globally, which will have a further impact on the quality of life (QOL) for presbyopes [3].

In people with both presbyopia and myopia, adequate correction for near and far vision is crucial for daily activities. The negative impact of presbyopia on both visual functions and QOL has been demonstrated with the use of questionnaires [5,6]. Similarly, the detrimental impact of myopia on vision-related outcomes has been shown in previous studies [7,8,9]. However, there is a lack of studies that look at the collective impact of myopia and presbyopia on QOL, the correction habits of presbyopic patients, and the impact of the combination of corrections utilised on their QOL. The refractive status and vision profile (RSVP) questionnaire and visual analogue scale (VAS) are methods that are well established, validated, and can be used to capture outcomes that cannot be measured through objective clinical assessments, enabling better management of the clinical practice and research evaluation of new treatments [10,11,12,13,14,15].

Therefore, this study aims to evaluate the QOL of presbyopes with low and high myopia and to determine how different optical corrections, namely progressive addition lenses (PAL) and single-vision distance (SVD) lenses, affect the QOL outcomes of presbyopes with various myopia severity.

## 2. Materials and Methods

### 2.1. Study Population

A total of 149 people aged 41 to 75 years, who had both myopia and presbyopia, participated in this study in the period between August 2016 and March 2018. Presbyopia was defined as the need for reading glasses, near addition, or, in some cases, removing the distance correction. Myopia was defined as spherical equivalent (SE) of ≤−0.50 diopters (D). All participants were myopic, with no more than 2.00 D of astigmatism in either eye; with anisometropia of less than 1.50 D; and with no history of any eye diseases (such as cataracts, glaucoma, age-related macular degeneration, or other eye complications) or surgeries.

The study adhered to the tenets of the Declaration of Helsinki, and ethics approval was obtained from the Singapore Polytechnic Ethics Review Committee. All tests were administered at the Singapore Polytechnic Optometry Centre after obtaining written informed consent from all participants.

### 2.2. Examinations

Only participants with distance visual acuity of at least 0.3 LogMAR (measured using the Early Treatment Diabetic Retinopathy Study (ETDRS) chart), near visual acuity of at least N5 (using the N-point near chart), and at least 1.9 log contrast sensitivity (using the Pelli-Robson Contrast Sensitivity chart with habitual correction) were included in the study. Study participants did not undergo any refraction assessment; thus, their distant spectacle power was used as the refractive error. Spectacle power was measured with an automated focimeter (Topcon, CL.100; Topcon Corporation, Tokyo, Japan).

### 2.3. Questionnaires

A detailed questionnaire was used to collect demographic (age, gender, occupation), ocular, and general medical history from the participants for the purpose of screening. In total, 182 were screened before 149 were recruited. The questionnaire was administered by research staff and completed by the participants themselves.

The original RSVP questionnaire consists of 42 questions. Four questions were omitted, as they were contact lens-related and did not apply to our study objective. There were 38 questions on the different types of vision-dependent activities to assess the level of difficulty in performing daily activities (Table 1). The items used a five-point rating scale. The 38 items were divided into the following five subscales: perception (5 items), expectation (5 items), functionality (14 items), and visual symptoms (13 items).

The current state of health (1 item) was measured using the VAS. This is a measure of perception that ranges across a continuum of values. VAS is a horizontal line, 100 mm in length, anchored by a word descriptor at the end—in this case, the “worst imaginable health state” at zero, and the “best imaginable health state” at 100.

### 2.4. Statistical Analysis

A Rasch analysis was used to transform the data, and, for further analysis, we used the Andrich rating scale model, with Winsteps software, version 3.68; (Winsteps, Chicago, IL, USA) [16,17]. The transformed scores were scaled from 0 to 100, with a higher score indicating better satisfaction and better QOL. Rasch analysis uses the raw score from the questionnaire and expresses the respondent’s outcome on a linear scale, which accounts for the unequal difficulties across all test items. The Rasch analysis was done for the overall QOL and the five subscales of perception, expectation, functionality, symptoms, and problems with glasses.

A chi-square test was used to test for differences in the proportion of participants between groups, and an analysis of variance (ANOVA) was used for the difference in the mean QOL between the groups, using statistical software Statistica, version 13.2 (TIBCO Software Inc., Palo Alto, CA, USA). Values of *p* < 0.05 were taken to be statistically significant differences.

## 3. Results

Of the 149 participants with both myopia and presbyopia, 108 (72.5%) were presbyopic with low myopia (SE ≤ −0.50 D to SE > −5.00 D), and 41 (27.5%) were presbyopic with high myopia (SE ≤ −5.00 D), with a mean age (±SD) of 52.1 ± 6.9 years. Moreover, 89 (59.7%) of the participants were females, and 60 (40.3%) were males. There was a significant difference in the distribution of gender, especially in the highly myopic group (*p* = 0.04). This difference in gender distribution did not have any effect on the QOL score (F (1, 145) = 0.30; *p* = 0.49), even with the additional effect among the myope group (F (1, 145) = 0.79; *p* = 0.32). Of the 41 with high myopia, most (85%) had an SE in the range of −5.00 D to −9.00 D. The power of the study was 99.4%, with an effect size of 0.83.

Eighty (53.7%) wore PALs, 61 (40.9%) wore SVD lenses, one (0.7%) wore single-vision near lenses, four (2.7%) wore bifocals, and the remaining three (2%) did not wear glasses. Among those with high myopia, 14 (35.9%) were SVD wearers, and 25 (64.1%) were PAL wearers. Among those with low myopia, 47 (46.1%) were SVD wearers, and 55 (53.9%) were PAL wearers. There were more females in the low-myopic group than in the high-myopic group (*p* = 0.04), and the distance-corrected habitual visual acuity was significantly better in the low-myopic group (−0.09 ± 0.09 logMAR) compared to the high-myopic group (−0.04 ± 0.09 logMAR) (*p* = 0.003) (Table 2).

The health-state score was significantly correlated with the QOL score, but the correlation was weak, r^2^ = 0.10 (*p* < 0.05). The health-state score was similar between the two myopic groups (*p* = 0.43), and between the PAL- and SVD-lens wearers (*p* = 0.81).

High myopes had a significantly lower overall QOL (51.7) than low myopes (65.2) (*p* < 0.001; Figure 1). High myopes also had significantly poorer functionality with glasses, with a score of 29.1, compared with those of presbyopes with low myopia, with a score of 36.2 (*p* = 0.01). Similarly, presbyopes with high myopia had poorer functionality without glasses (49.4) than low myopes (57.0; *p* = 0.04).

With glasses, a greater proportion of high myopes had difficulty driving at night (low myopes 27.5% vs. high myopes 54.2%) (*X*^2^ (1, *n* = 93) = 5.6; *p* = 0.02) and driving in the rain (11.3% low myopes vs. 36.7% high myopes; *X*^2^ (1, *n* = 92) = 8.3; *p* = 0.004). High myopes also had more issues swimming with correction (19.4% low myopes vs. 36.7% high myopes; *X*^2^ (1, *n* = 92) = 4.5; *p* = 0.03). Without glasses, high myopes had greater difficulty reading and doing near work (42% low myopes vs. 83.9% high myopes; *X*^2^ (1, *n* = 131) = 16.6; *p* < 0.001). They were also less able to see clearly when they woke up (42.3% vs. 88.2%; *X*^2^ (1, *n* = 131) = 21.4; *p* < 0.001) or see the alarm clock (40.4% vs. 80.0%; *X*^2^ (1, *n* = 129) = 16.0; *p* < 0.001).

For presbyopes with low myopia, the group using PAL had significantly better overall QOL than SVD lens users (*p* = 0.04; Figure 2), although there was no significant difference between SVD lens and PAL wearers in all the other subscales, such as perception, expectation, functionality, and symptoms.

In the group of presbyopes with high myopia, those who wore PAL had significantly better perception (61.2) than those who wore SVD lenses (46.9; *p* = 0.03). High myopes wearing SVD lenses stated that they were more often afraid to do things because of their vision (SVD 57.1% vs. PAL 28%; *X*^2^ (1, *n* = 39) = 4.3; *p* = 0.04), and were more frustrated with their glasses (71.4% vs. 32.0%; *X*^2^ (1, *n* = 39) = 5.6; *p* = 0.02). No other significant differences were found for the other subscales, for presbyopes with high myopia wearing PAL and SVD lenses (Figure 3).

## 4. Discussion

### 4.1. Significant Findings

Presbyopes with high myopia had poorer overall QOL compared to those with low myopia. Similarly, high myopes had worse functionality scores compared to low myopes. Compared to SVD users, PAL users, on average, had better overall QOL scores for both myopic groups. PAL users also had better perception scores for high myopes. The difference in gender distribution did not have a significant effect on the QOL score.

The highly myopic group had significantly poorer visual acuity, with a difference of 0.05 logMAR, which equates to 2–3 letters from the visual acuity chart. This slight decrease in visual acuity may not be considered clinically significant by clinicians. However, it may have a tangible effect, contributing to poorer QOL and functionality outcomes with glasses. Therefore, this study’s outcome from the QOL reflected the tangible effects of reduced vision felt by the participants, which were often dismissed as insignificant by clinicians.

Reduced best-corrected visual acuity with spectacle lenses in high myopia has been found in previous studies [9,18,19,20,21,22,23,24,25]. In addition, there was a higher proportion of high myopes who experienced severe trouble with driving at night and in the rain [26]. Besides visual acuity affecting the corrected vision of high myopes, the night vision threshold [26], higher-order aberration [20,21], and larger pupil size may also contribute to poorer vision under dim lighting, as experienced when driving at night and in the rain. Further physiological stretching from axial elongation due to myopia also reduces the function and resolution of photoreceptors [22,26]. Some studies also found reduced contrast sensitivity at high spatial frequencies in fully corrected high myopes, which may contribute to reduced functionality with glasses [27]. However, we did not find any differences in contrast sensitivity between low and high myopic groups, as found by Collins et al. [19]. Further investigation is required to measure contrast sensitivity at different spatial frequencies in order to elucidate the underlying cause of reduced functionality with glasses.

It was expected that the difference in the refractive error between high- and low- myopic groups (−5.52 ± 2.4 D compared to −3.1 ± 1.7 D; *p* < 0.001) would have a significant impact on the unaided visual acuity of high-myopic groups, even though it was not measured. With significantly poorer vision without glasses, a higher proportion of high-myopic presbyopes would have issues seeing both far and near, as they are severely under-corrected for both distances. This would result in a poorer outcome in functionality without glasses for the high-myopic group. The poorer outcome in QOL regarding uncorrected vision was also reflected in other studies [18,28,29]. Our study shows that a larger proportion of high myopes had difficulty reading and doing near work, as well as waking up with clear vision and looking at an alarm clock without glasses. All the affected activities, as mentioned above, were near-distance activities, as also reported in other studies [18,29]. The lack of distance activities reported without glasses was due to the inability to carry them out without glasses. No high myopes drove without glasses.

This study found that highly myopic PAL wearers had a better score for perception subscales compared to SVD lens wearers. In the perception subscale, highly myopic SVD lens wearers were more “afraid to do things due to their vision” and were also more “frustrated with their glasses.” SVD lenses only correct distance vision and not near vision; hence, highly myopic SVD lens wearers will have poor near and intermediate vision, with or without glasses. To overcome blurred vision due to working distance, they may need to remove and put on SVD glasses more frequently, adding to the frustration. Compared with SVD wearers, low-myopic PAL wearers also had a significantly better overall QOL, with no other difference in the other subscales. Despite the lack of a significant difference in each subscale, the significant differences in the overall QOL may be due to the additive effect of multiple components. Other studies have found that near vision is affected while using SVD lenses for presbyopes, while having better outcomes using PAL [30,31,32,33]. Poorer near and intermediate vision with SVD lenses may significantly affect QOL outcomes in low myopes; they may also significantly affect the perception subscale for high myopes. Moreover, Pesudovs et al., 2006 found that PAL wearers have reduced sensitivity to light, eye pain, and redness compared to SVD lens wearers, while doing near work, for early presbyopes [33]. As such, the visual comfort from PAL could be another factor in this outcome.

### 4.2. Strengths and Limitations of the Study

This is the first study that explores the correction habits of presbyopes and the impact of the severity of myopia on QOL. This study was able to measure the subjective differences between the severity of myopia and the types of visual correction, which was otherwise not significantly different from clinical measures. However, the recruitment rate of patients with high myopia (27.5%) was much lower compared with those having low myopia (72.5%). This, however, is a reflection of myopia’s prevalence in the population [3]. Refraction and axial length measurements were not conducted to directly link the causal effect of refractive error and elongation of the eye to the QOL outcome. Unaided visual acuity and contrast sensitivity with spatial frequencies need to be measured to directly understand the contribution of these factors to some of the subscales, such as functionality with and without glasses. More details such as the lens design of PAL should be included in order to further understand whether it has an impact on QOL. Though the RSVP questionnaire has been shown to be deficient in several psychometric properties with underutilised response scales, it was chosen not only because it was validated but also because it includes measures for quality of vision and life [11,12,13,14,15,34,35,36].

### 4.3. Suggestions for Future Work

From this study, the QOL assessment recorded outcomes that could not be measured through typical clinical tests or may be deemed clinically insignificant. Hence, such questionnaires should be administered during dispensing to achieve higher success rates. Work should be done to understand which are the important and contributing subscales for each eye condition and interventions, in order to apply the right questionnaire for each condition. A systematic review could be done on all types of vision correction used for presbyopia, such as PAL, SVD, contact lenses, and intraocular lenses, in order to understand their impact on QOL.

## 5. Conclusions

This study was able to measure significant subjective feedback from the RSVP questionnaire that was not found clinically (visual acuity). It was found that a significantly higher proportion of highly myopic presbyopes reported lower vision-related QOL across both the QOL and functionality subscales. Despite having similar best-corrected vision, PAL wearers had better QOL outcomes than SVD lens wearers with low myopia. Moreover, PAL wearers with high myopia had better perception than SVD wearers with high myopia. Therefore, wearing PAL could be a better option to improve the QOL in myopic presbyopes of various myopia levels.

## Figures and Tables

**Figure 1 jcm-10-01589-f001:**
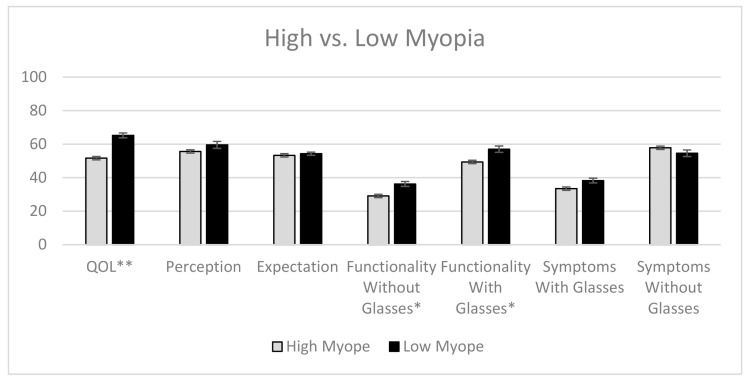
The quality of life (QOL) scores of presbyopes with low myopia versus high myopia, with error bars representing standard error. * Statistically significant, with a *p*-value of <0.05; ** statistically significant, with a *p*-value of <0.001.

**Figure 2 jcm-10-01589-f002:**
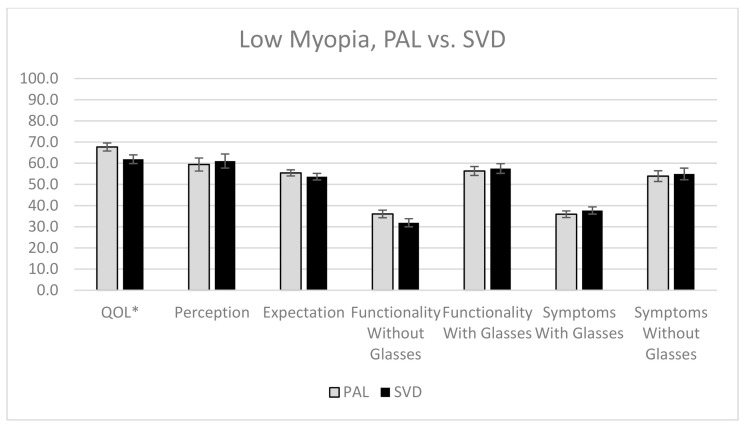
The QOL scores of presbyopes with low myopia who wore PAL and SVD lenses for all five subscales, with error bars representing standard error. * Statistically significant, with *p*-value of <0.05.

**Figure 3 jcm-10-01589-f003:**
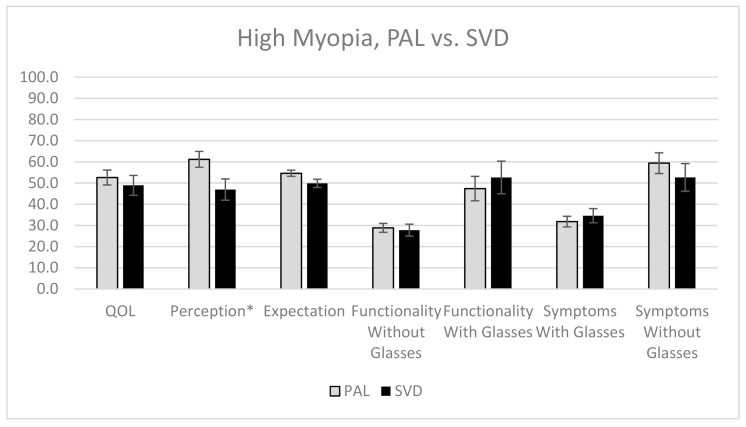
The QOL scores of highly myopic presbyopes who wore PAL and SVD lenses for all five subscales, with error bars representing standard error. * Statistically significant, with *p*-value of <0.05.

**Table 1 jcm-10-01589-t001:** Summary of the refractive status and vision profile (RSVP) questionnaire.

Questions	Scale
Perception (1) I worry about my vision. (2) My vision holds me back. (3) I am frustrated with my vision. (4) My vision makes me less self-sufficient. (5) Because of my vision, there are things I am afraid to do.	(1) Never (2) Rarely (3) Sometimes (4) Often (5) Always
Expectation (1) I am frustrated to use glasses to get the best possible vision. (2) I could accept less than perfect vision if I didn’t need glasses any more. (3) As long as I could see well enough to drive without wearing glasses, I wouldn’t mind having a vision that was less than perfect. (4) I am only satisfied with my life if I have very sharp vision without glasses. (5) I think my vision will be worse in the future.	(1) Strongly disagree (2) Disagree (3) Neither agree nor disagree (4) Agree (5) Strongly agree
Functionality (With and without correction) (1) Watching TV or movies (2) Working or outdoor activities (3) Taking care of or playing with children (4) Seeing your alarm clock (5) Seeing clearly when you wake up (6) Seeing a clock on the wall (7) Doing your job (8) Doing sports/recreation (9) Swimming (10) Your social life (11) Reading and near work (12) Driving at night (13) Driving in the rain (14) Driving when there is a glare from oncoming headlights	(1) Not applicable (2) No difficulty at all (3) A little difficulty (4) Moderate difficulty (5) Severe difficulty (6) So much difficulty that I did not do the activity with this type of correction
Visual Symptoms (With and without correction) (1) Your eyes feeling irritated (2) Drafts (from heating or air-conditioning) blowing into your eyes (3) Eyes being sensitive to light (4) Pain in your eyes (5) Changes in your vision during the day (6) Your vision is cloudy or foggy (7) Glare (reflections off shiny surfaces, snow) (8) Things looking different out of one eye versus the other (9) Seeing a halo around lights (10) Seeing in dim light (11) Your depth perception (12) Things appearing distorted (13) Judging distance when going up or down steps (stairs, curbs)	
The current state of healthYour own health state today	0 (worst imaginable health state) to 100 (best imaginable health state) using a visual analogue scale

**Table 2 jcm-10-01589-t002:** Baseline characteristics of presbyopic participants with low myopia and high myopia (*n* = 149). Mean ± standard deviation. PAL, progression addition lens; SVD, single-vision distance lens; VAS, visual analogue scale. * Statistically significant with *p*-value of <0.05.

	Low Myopia (n = 108)	High Myopia (n = 41)	*p*-Value
Mean age (years)	51.8 ± 6.6	52.8 ± 7.7	0.43
Gender, *n* (%)			
Female	70 (64.8%)	19 (46.3%)	0.04 *
Male	38 (35.2%)	22 (53.7%)	
Mean spherical equivalent, diopters	−3.1 ± 1.7	−5.6 ± 2.4	<0.001 *
Type of glasses, *n* (%)			
PAL	55 (50.9%)	25 (61.0%)	0.40
SVD	47 (43.5%)	14 (34.1%)	0.50
Others	6 (5.6%)	2 (4.9%)	-
Mean distance visual acuity, LogMAR	−0.09 ± 0.09	−0.04 ± 0.09	0.003 *
Mean near visual acuity, LogMAR	0.25 ± 0.20	0.25 ± 0.14	0.57
Mean log contrast sensitivity	1.94 ± 0.02	1.94 ± 0.03	0.22
Current health (VAS)	77.48 ± 1.50	73.07 ± 2.42	0.12

## Data Availability

Please contact corresponding author for data.

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
