# Peer review of "Quality of Life in Presbyopes with Low and High Myopia Using Single-Vision and Progressive-Lens Correction"

_jcm, 2021, doi:10.3390/jcm10081589_

Round 1

Reviewer 1 Report

The authors aims to evaluate the impact of severity of myopia and the type of visual correction in presbyopia on vision-related quality of life (QOL) using the Refractive Status and Vision Profile (RSVP) questionnaire and found that High myopic PAL users had a better perception outcome than SVD lens wearers. Low myopic PAL wearers had a better QOL than SVD wearers.

Major concerns

#1 The questionnaire is validated?

#2 The interested topic is very limited.

#3 Statistically significant different were very low

#4 Reference cite does not match with the JCM instructions.

#5 Figure 1 , express count in percentage, the figure is unnecessary.

#7 The manuscript has flaws in method section.

#8 The scientific question seems to be poor.

#9 Conclusions were weak.

#10 Some references were old and out of JCR

Author Response

Comments from Reviewer #1:

The authors aims to evaluate the impact of severity of myopia and the type of visual correction in presbyopia on vision-related quality of life (QOL) using the Refractive Status and Vision Profile (RSVP) questionnaire and found that High myopic PAL users had a better perception outcome than SVD lens wearers. Low myopic PAL wearers had a better QOL than SVD wearers.

Major concerns

#1 The questionnaire is validated?

Author response: Thank you for the comment. The Refractive Status and Vision Profile (RSVP) questionnaire had been validated by previous studies as cited below.

  1. Vitale, S. The Refractive Status and Vision Profile A Questionnaire to Measure Vision-Related Quality of Life in Persons with Refractive Error. Ophthalmology 2000, 107 (8), 1529–1539.
  2. Nichols, J. J. Reliability and Validity of Refractive Error–Specific Quality-of-Life Instruments. Archives of Ophthalmology 2003, 121 (9), 1289.
  3. Khadka, J.; McAlinden, C.; Pesudovs, K. Quality Assessment of Ophthalmic Questionnaires. Optometry and Vision Science 2013, 90 (8), 720–744.
  4. Wu, X. Y.; Ohinmaa, A.; Johnson, J. A.; Veugelers, P. J. Assessment of Children’s Own Health Status Using Visual Analogue Scale and Descriptive System of the EQ-5D-Y: Linkage between Two Systems. Quality of Life Research 2013, 23 (2), 393–402.

We will include these citations in the Introduction (Line 48-51), “The refractive status and vision profile (RSVP) questionnaire and visual analogue scale (VAS) are methods that are well established, validated and can be used to capture outcomes that cannot be measured through objective clinical assessments, enabling better management of the clinical practice and research evaluation of new treatments.”

#2 The interested topic is very limited.

Author response: Thank you for the comment. There is currently no existing publication about the QOL of myopic presbyopes using progressive addition lenses (PAL) and single vision lenses to correct their vision. It is important to report QOL outcomes on top of visual acuity performance, which would be helpful for clinicians to communicate the added benefits of PAL in improving QOL.

#3 Statistically significant different were very low

Author response: Thank you for the comment. We agree that the statistical significance is low, however, it is statistically significant as the power of the study was 99.4% with an effect size of 0.83. We have indicated this outcome in Results (Line 117-118), “The power of the study was 99.4% with an effect size of 0.83.”

#4 Reference cite does not match with the JCM instructions.

Author response: Thank you for the advice. We have edit the whole reference list in accordance with the JCM requirements.

#5 Figure 1 , express count in percentage, the figure is unnecessary.

Author response: Thank you for the advice. We have now removed Figure 1 and expressed the distribution of low and high myopes in the text under Results.

#7 The manuscript has flaws in method section.

Author response: Thank you for the comment. We have now double-checked the methods section and made changes below.

2.1 Study Population (Line 61), A total of 149 people, aged 41 to 75 years, who had both myopia and presbyopia participated in this study in the period between August 2016 to March 2018. “

2.2 Examinations (Line 72-76), “Only participants with distance visual acuity of at least 0.3 LogMAR measured using the Early Treatment Diabetic Retinopathy Study (EDTRS) chart, near visual acuity of at least N5 using the N-point near chart and at least 1.9 log contrast sensitivity using the Pelli-Robson Contrast Sensitivity chart with habitual correction were included in the study. “

2.3 Questionnaires (Line 82-84), “The questionnaire was administered by research staff and filled up by the participants themselves.

2.3 Questionnaires (Line 91-94), “The current state of health (1 item) was measured using the VAS. It is a measure of perception that range across a continuum of values. VAS is a horizontal line, 100mm in length, anchored by word descriptor at the end. In this case, “Worst imaginable health state” at zero and “Best imaginable health state” at 100.”

#8 The scientific question seems to be poor.

Author response: Thank you for the comment. For better clarity, we have now rephrased our scientific question in Introduction (Line 52-55), “ this study aims to evaluate the QOL of presbyopes with low and high myopia, and determine how different optical corrections, namely PAL and SVD, affect the QOL outcomes of presbyopes with various myopia severity”.

#9 Conclusions were weak.

Author response: Thank you for the comment. We have now reorganized our conclusions for better clarity.

Conclusion (Line 256-263), “This study was able to measure significant subjective feedback from RSVP questionnaire that was not found clinically (visual acuity). It was found that a significantly higher proportion of highly myopic presbyopes reported lower vision-related QOL across both QOL and functionality subscales. Despite having a similar best-corrected vision, PAL wearers had better QOL outcome than SVD lens wearers with low myopia, and PAL wearers with high myopia had better perception than SVD wearers with high myopia. Therefore, wearing PAL could be a better option to improve the QOL in myopic presbyopes of various myopia levels.”

#10 Some references were old and out of JCR

Author response: Thank you for your advice. Yes, we agree that some journals related to QOL in presbyopia were not on the JCR as such publication is really not common. As such, it is important to publish more in this area so that there is more information about managing presbyopia with different levels of myopia and optical corrections other than surgical interventions like IOLs.

Reviewer 2 Report

I would like to congratulate the authors for this interesting research which is very needed. There are many many papers about QOL impact of presbyopia correction with modern IOLs, but very few research about the impact of QOL of using progressive ophthalmic lenses. Thank you!!!

Only minor comments:

1.- I miss all the information about the optical characteristics of the lenses adapted. This information is necessary to complement the paper.

2.- Were the questionnaired filled out by patients or the practitioner? This is a relevant issue that should be clarified.

3.- Please add some comparison with the QOL outcomes reported with multifocal IOLs and contact lenses

Author Response

Comments from Reviewer #2

I would like to congratulate the authors for this interesting research which is very needed. There are many many papers about QOL impact of presbyopia correction with modern IOLs, but very few research about the impact of QOL of using progressive ophthalmic lenses. Thank you!!!

Only minor comments:

1.- I miss all the information about the optical characteristics of the lenses adapted. This information is necessary to complement the paper.

Author response: Thank you for the advice. The participants were wearing their own spectacle lenses, thus details on the optical characteristics, brand and model of lenses were not documented at that point of the study. We only documented information on the type of lenses that they were wearing: progressive addition lenses (PAL), single vision lenses for distance (SVD), for near (SVN) and bifocals. From which, the majority of the participants wore PAL and SVD, hence we focused on reporting the differences between these 2 groups.

We have now included the lack of data collection of the specific optical characteristics of the lenses worn by the participants as a limitation in our study under 4.2 Strengths and Limitations of the study (Line 242-243), More details like the lens design of PAL should be included to further understand if it has an impact on QOL.”

2.- Were the questionnaired filled out by patients or the practitioner? This is a relevant issue that should be clarified.

Author response: Thank you for the query. The RSVP questionnaire was filled up by the participants. We have amended the manuscript 2.3 Questionnaire (Line 82-84), “The questionnaire was administered by trained research staff and filled up by the participants themselves to ensure reliability and accuracy of responses.”

3.- Please add some comparison with the QOL outcomes reported with multifocal IOLs and contact lenses

Author response: Thank you for the suggestion. We were not able to do the comparison with contact lenses and multifocal IOLs as the study objective and questionnaire used were different. This could be done in the future as a systematic review. Nevertheless, we included in 4.3 Suggestions for future work (Line 252-254), “A systematic review could be done on all the vision correction used for presbyopia like PAL, SVD, contact lenses and intraocular lenses to understand their impact on QOL.

Reviewer 3 Report

This study attempts to evaluate the impact of severity of myopia and the type of visual correction in presbyopia on vision-related quality of life (QOL) using the Refractive Status and Vision Profile (RSVP) questionnaire. 

Prior to any consideration for publication, this study should address the following:

Comments of major importance:

  • The study does not include any power analysis that could determine if the sample size is adequate.
  • The phrase "Rasch analysis was performed on five subscales: perception, expectation, functionality, symptoms, and problems with glasses." in the abstract is not explained in the main manuscript. The outcomes derived from Rasch analysis could be better described in the Results section. 

Comments of minor importance:

- Abstract / line 20, 21: Please replace the phrase : "wore progressive addition lenses in the high myope group (PAL)" with the phrase "wore progressive addition lenses (PAL) in the high myope group"

- p. 1 / lines 29-31 / "Quality of life (QOL).... visual analogue scales (VAS).": Abbreviations should be defined in parentheses the first time they appear in the main text and used consistently thereafter. Therefore, the definitions at the beginning of the manuscript could be omitted and they should be defined in the main manuscript (eg. p. 2, line 79: RSVP, p. 4, line 110: PAL, SVD).

In specific, it would be better if the terms SVD and PAL were first mentioned and described in the Introduction section.

Results:

- p. 4 / line 103 / "60 (57.4%) were males ": Do you mean 60 (40,3%) were males"?

- Abbreviations should also be explained in the figures of the manuscript. For instance, QOL in Fig 2, 3, 4 and PAL & SVD in Fig 3, 4.

- The following reference should be added: "Vitale S, Schein OD, Meinert CL, Steinberg EP. The refractive status and vision profile: a questionnaire to measure vision-related quality of life in persons with refractive error. Ophthalmology. 2000 Aug;107(8):1529-39."

Author Response

Comments from Reviewer #3

This study attempts to evaluate the impact of severity of myopia and the type of visual correction in presbyopia on vision-related quality of life (QOL) using the Refractive Status and Vision Profile (RSVP) questionnaire.

Prior to any consideration for publication, this study should address the following:

Comments of major importance:

  • The study does not include any power analysis that could determine if the sample size is adequate.
  • The phrase "Rasch analysis was performed on five subscales: perception, expectation, functionality, symptoms, and problems with glasses." in the abstract is not explained in the main manuscript. The outcomes derived from Rasch analysis could be better described in the Results section.

Author response: Thank you for the comments. We did a power analysis and the outcome using mean QOL score between high and low myope group showed that the power of the study was 99.4% with effect size of 0.83. This shows that the sample size for this study was more than adequate. We have included this information in Results (Line 117-118), “The power of the study was 99.4% with an effect size of 0.83.”

With regards to the Rasch analysis for each subscales, we have included the explanation in 2.4 Statistical Analysis (Line 103-104),Rasch analysis was done for the overall QOL and five subscales: perception, expectation, functionality, symptoms, and problems with glasses.”

Comments of minor importance:

- Abstract / line 20, 21: Please replace the phrase : "wore progressive addition lenses in the high myope group (PAL)" with the phrase "wore progressive addition lenses (PAL) in the high myope group"

Author response: Thank you for your suggestion. We made the change at Abstract (Line 20), "wore progressive addition lenses (PAL) in the high myope group".

- p. 1 / lines 29-31 / "Quality of life (QOL).... visual analogue scales (VAS).": Abbreviations should be defined in parentheses the first time they appear in the main text and used consistently thereafter. Therefore, the definitions at the beginning of the manuscript could be omitted and they should be defined in the main manuscript (eg. p. 2, line 79: RSVP, p. 4, line 110: PAL, SVD).

In specific, it would be better if the terms SVD and PAL were first mentioned and described in the Introduction section.

Author response: Thank you for the comment. We have now replaced the abbreviations accordingly and double-checked the manuscript for such instances.

Results:

- p. 4 / line 103 / "60 (57.4%) were males ": Do you mean 60 (40,3%) were males"?

Author response: Thank you for the comment. We have now changed the typo error to 60 (40.3%) and double-checked the results.  The changes are at Results (Line 113), “...and 60 (40.3%) were males.”

- Abbreviations should also be explained in the figures of the manuscript. For instance, QOL in Fig 2, 3, 4 and PAL & SVD in Fig 3, 4.

Author response: Thank you for the suggestion. We have now included the abbreviations for all the figures of the manuscript.

- The following reference should be added: "Vitale S, Schein OD, Meinert CL, Steinberg EP. The refractive status and vision profile: a questionnaire to measure vision-related quality of life in persons with refractive error. Ophthalmology. 2000 Aug;107(8):1529-39."

Author response: Thank you for the suggestion. We have now added this reference in the Introduction (Line 51) and 4.2 Strengths and Limitations of the study (Line 245).

Reviewer 4 Report

General comments

The concept of this study is great and this type of data needs to be published and recognized by the profession.

 The manuscript does not contain a reference for the source of the RSVP questionnaire. This is important to know because of how it was developed and what it actually measures. In follow-up evaluations, the RSVP was found to have some limitations and inadequacies and this should be noted.

This is from the RSVP study “ Our primary goal was to assess the properties of the RSVP in a population representative of those considering refractive surgery. “ and “ Our study design does not allow a rigorous assessment of representativeness. “  Also, the RSVP was developed in a pre-presbyopic group with a mean age of 37 years, much different than this study group. These limitations should be mentioned in the discussion.

Khadka et al. Quality Assessment of Ophthalmic Questionnaires: Review and Recommendations. Optom Vis Sci. 2013;90:720-744.

Vitale et al. The Refractive Status and Vision Profile. Ophthalmology 2000;107:1529–1539.

The same comment applies to the VAS that was not described.

You should note that there were more than double the number of low myopes and high myopes in the study. Comparisons between the groups may be limited or swayed by these numbers.

Throughout the paper, you use both “QOL” and “vision-related QOL”. Those are two different things. Please go through and modify the usage of those terms to accurately reflect the point you are making. Be careful how those are worded and look at the RSVP to verify exactly what it measures.

Specific comments

In section 2.1, change the word “case” to “cases”

Author Response

Comments from Reviewer #4

General comments

The concept of this study is great and this type of data needs to be published and recognized by the profession.

 The manuscript does not contain a reference for the source of the RSVP questionnaire. This is important to know because of how it was developed and what it actually measures. In follow-up evaluations, the RSVP was found to have some limitations and inadequacies and this should be noted.

This is from the RSVP study “ Our primary goal was to assess the properties of the RSVP in a population representative of those considering refractive surgery. “ and “ Our study design does not allow a rigorous assessment of representativeness. “  Also, the RSVP was developed in a pre-presbyopic group with a mean age of 37 years, much different than this study group. These limitations should be mentioned in the discussion.

Khadka et al. Quality Assessment of Ophthalmic Questionnaires: Review and Recommendations. Optom Vis Sci. 2013;90:720-744.

Vitale et al. The Refractive Status and Vision Profile. Ophthalmology 2000;107:1529–1539.

The same comment applies to the VAS that was not described.

Author response: Thank you for the suggestion. We have now added the references below for the RSVP and VAS questionnaires in the Introduction (Line 51) and 4.2 Strengths and Limitations of the study (Line 245).

  1. Vitale, S. The Refractive Status and Vision Profile A Questionnaire to Measure Vision-Related Quality of Life in Persons with Refractive Error. Ophthalmology 2000, 107 (8), 1529–1539.
  2. Nichols, J. J. Reliability and Validity of Refractive Error–Specific Quality-of-Life Instruments. Archives of Ophthalmology 2003, 121 (9), 1289.
  3. Khadka, J.; McAlinden, C.; Pesudovs, K. Quality Assessment of Ophthalmic Questionnaires. Optometry and Vision Science 2013, 90 (8), 720–744.
  4. Wu, X. Y.; Ohinmaa, A.; Johnson, J. A.; Veugelers, P. J. Assessment of Children’s Own Health Status Using Visual Analogue Scale and Descriptive System of the EQ-5D-Y: Linkage between Two Systems. Quality of Life Research 2013, 23 (2), 393–402.

You should note that there were more than double the number of low myopes and high myopes in the study. Comparisons between the groups may be limited or swayed by these numbers.

Author response: We agree with the reviewer that the difference in numbers between low myopes and high myopes were large in our study, thus there is a possibility that our data may be skewed. Other studies such as the Singapore Malay Eye Study (SiMES) had also shown lower recruitment levels for high myopes (27.8%) as compared to low myopes (72.2%) which showed more than double the number of low myopes than high myopes. In addition, a study by Brien A.Holden mentioned that by 2050, the ratio of low myopes to high myopes would be 5:1. If our study recruited more than five times the numbers of low myopes compared to high myopes, we would be more concerned about the validity of the results. Having said that, we had previously noted this as a limitation in our study under 4.2 Strengths and Limitations of the study (Line 223-225), However, the recruitment rate of patients with high myopia (27.5%) was much lower compared to low myopia (72.5%). This, however, is a reflection of myopia prevalence in the population.”.

Lamoureux, E. L.; Wang, J.; Aung, T.; Saw, S. M.; Wong, T. Y.; Myopia and Quality of Life: The Singapore Malay Eye Study (SiMES). Invest. Ophthalmol. Vis. Sci. 2008,49(13),4469.

Throughout the paper, you use both “QOL” and “vision-related QOL”. Those are two different things. Please go through and modify the usage of those terms to accurately reflect the point you are making. Be careful how those are worded and look at the RSVP to verify exactly what it measures.

Author response: Thank you for your comments. We made the changes by removing vision related QOL to just using QOL in the manuscript to avoid confusion. Introduction (Line53), “The refractive status and vision profile (RSVP) questionnaire and visual analogue scale (VAS) …”

Specific comments

In section 2.1, change the word “case” to “cases”

Author response: Thank you for the edit. We made the change asked by the reviewer at 2.1 Study Population (Line 62), “...in some cases…”

Round 2

Reviewer 1 Report

Comments solved